# StanDep: Capturing transcriptomic variability improves context-specific metabolic models

**Chintan J. Joshi**[1], **Song-Min Schinn**[1], **Anne Richelle**[1], **Isaac Shamie**[2,3], **Eyleen J. O'Rourke**[4], **Nathan E. Lewis**[1,2,5]*

**1** Department of Pediatrics, University of California, San Diego, School of Medicine, La Jolla, CA, United States of America, **2** Novo Nordisk Foundation Center for Biosustainability at the University of California, San Diego, School of Medicine, La Jolla, CA, United States of America, **3** Bioinformatics and Systems Biology Program, University of California, San Diego, United States of America, **4** Department of Biology, University of Virginia, Charlottesville, VA, United States of America, **5** Department of Bioengineering, University of California, San Diego, La Jolla, CA, United States of America

* nlewisres@ucsd.edu

**Data Availability Statement:** All relevant data are within the manuscript and its Supporting Information files. All StanDep (MATLAB files) files

## Abstract

Diverse algorithms can integrate transcriptomics with genome-scale metabolic models (GEMs) to build context-specific metabolic models. These algorithms require identification of a list of high confidence (core) reactions from transcriptomics, but parameters related to identification of core reactions, such as thresholding of expression profiles, can significantly change model content. Importantly, current thresholding approaches are burdened with setting singular arbitrary thresholds for all genes; thus, resulting in removal of enzymes needed in small amounts and even many housekeeping genes. Here, we describe StanDep, a novel heuristic method for using transcriptomics to identify core reactions prior to building context-specific metabolic models. StanDep clusters gene expression data based on their expression pattern across different contexts and determines thresholds for each cluster using data-dependent statistics, specifically standard deviation and mean. To demonstrate the use of StanDep, we built hundreds of models for the NCI-60 cancer cell lines. These models successfully increased the inclusion of housekeeping reactions, which are often lost in models built using standard thresholding approaches. Further, StanDep also provided a transcriptomic explanation for inclusion of lowly expressed reactions that were otherwise only supported by model extraction methods. Our study also provides novel insights into how cells may deal with context-specific and ubiquitous functions. StanDep, as a MATLAB toolbox, is available at https://github.com/LewisLabUCSD/StanDep

## Author summary

Integration of transcriptomics data with genome-scale metabolic models is appealing but challenging due to the number of parametric decisions required to be made to by the user. This is further exacerbated by models failing to capture functionalities which are important for cellular maintenance. In this study, we propose a thresholding method for functionally qualifying a metabolic reaction to be active. We used our method to extract

are available from the Github (https://github.com/LewisLabUCSD/StanDep).

**Funding:** This work was supported by the NIGMS (grant no. R35 GM119850, NEL), a Lilly Innovation Fellows Award to CJ and AR, and funding from the Keck Foundation (EJOR). The funders had no role in study design, data collection and analysis, decision to publish, or preparation of the manuscript.

**Competing interests:** The authors have declared that no competing interests exist.

models of NCI-60 cancer cell lines, human tissues, and C. elegans cell types. We show that our thresholding method improves the coverage of functions required for cellular maintenance. We also validated and compared models built with our approach against those with existing approaches using CRISPR-Cas9 essentiality screens. Overall, our study provides novel insights into how cells may deal with context-specific and ubiquitous functions.

## Introduction

Integration of omics data with genome-scale metabolic models (GEMs) has facilitated insights into diverse questions, spanning from the elucidation of disease mechanisms [1,2] to the identification of drug targets [3–5]. Furthermore, a recent rise in comprehensively quantified omics data for many tissues and cell types [6–9] presents an opportunity to study context-specific behavior (i.e., tissue, cell type, environmental conditions, or other variations to which cells are exposed) [10–12]. Such studies depend on the omics-integrated models to include context-relevant genes and reactions. Unfortunately, due to over simplified assumptions of which genes are expressed or not, current omics-integration methods may fail to include important genes, leading to less-predictive models [13]. Here we present an improved method to identify context-relevant genes robustly, leading towards models that better describe context-specific metabolism.

The integration of omics data and GEMs is complicated by the complexity of cellular metabolism and enzyme regulation. Metabolic phenotypes are driven by not only gene expression alone but also other orthogonal processes, including enzyme assembly, post-translational modifications, localization, and substrate concentration. In other words, gene expression data provide considerable, but partial, insight into metabolic activity. To address this, data integration efforts often infer a 'core' set of active reactions from gene expression data. This 'core reaction set' is then used to produce a context-specific model via various model-extraction algorithms [14–20], which take into account network topology, model objective function or additional data (Fig 1, grey). Inference of the 'core reaction set' typically involves defining a gene expression level as a threshold parameter–genes expressed above the defined threshold are interpreted to be active and part of the 'core'. This threshold parameter has a large influence on the data integration process and its resulting model, according to recent systematic benchmarking studies [13,21].

Despite this importance, thresholds have often been poorly defined, and it is not fully understood how these thresholds should be set. Most often a single threshold value is used to evaluate all genes, disregarding complex and pathway-specific regulations over metabolism. Furthermore, such thresholds are often left to be defined by the user with little guidance or standardization, leading to varying and arbitrary model parameterization. Lastly, such single, catch-all thresholds often fail to identify lowly expressed but biologically important genes, including 'housekeeping' genes which are constitutively expressed for tissue maintenance functions [22]. A limited number of such housekeeping reactions can be 'rescued' by the model-extracting methods. A method that uses differential expression analysis to identify the core reaction set [23] will eliminate the arbitrariness involved in selection of thresholds; but may fail to capture such housekeeping reactions as it removes such features from the data. Specifically, housekeeping reactions involved in the central carbon metabolism or tied to the biomass objective function are particularly favored to be 'rescued' by these algorithms. Despite this, a sizeable portion of housekeeping genes are seldom included into the resulting model,

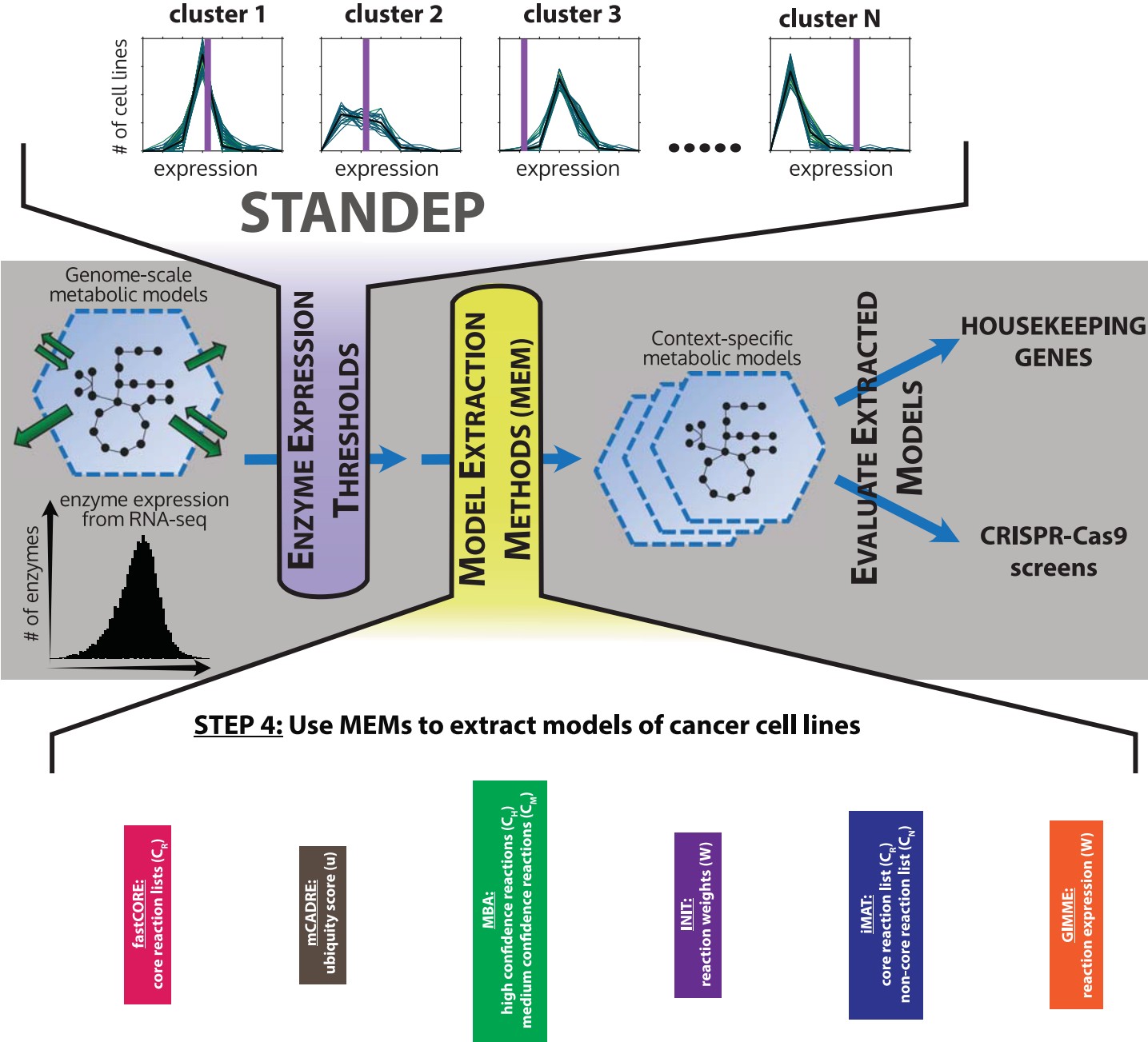

**Fig 1. Basic methodology of extracting context-specific metabolic models (CSMMs) using StanDep and model extraction methods (MEMs).** In grey panel, metabolic gene expression data is converted to enzyme expression (i.e. combined expression of enzyme-coding genes for each reaction) data which are subjected to thresholds using a given approach (for e.g. global, localT1, localT2, or StanDep) and tailored for a given MEM to extract and evaluate CSMMs. In purple extension, StanDep is applied to enzyme expression data by clustering enzyme expression values (Step 1) followed by subjecting each cluster to thresholds calculated by Eqs (1–4) (Step 2). This is then followed by tailoring the output of StanDep to integrate with different MEMs (Step 3). We note that enzyme-specific variability refers to the variability of combined

expression of enzyme-coding genes for each reaction. In yellow extension, the extracted CSMMs (Step 4) are then evaluated using housekeeping genes/reactions and gene essentiality screens.

preventing context-specific models from describing important cellular functions [13]. Such challenges could be addressed by a thresholding method that better captures the complex relationship between gene expression and metabolism.

Here we propose a novel thresholding approach called StanDep, which clusters gene expression data based on their expression pattern across conditions (Fig 1, purple). This enables genes that are expressed similarly to be interpreted together by common thresholds. In other words, StanDep better accounts for regulatory complexity via a group of heuristically derived thresholds rather than a single one-size-fits-all threshold. Using this finer-grained thresholding approach, StanDep captures more housekeeping genes into context-specific models compared to other thresholding methods for a wide variety of model-extraction algorithms. We further validated this method by predicting essential genes in cancer cells [24–27], and analyzing 32 human tissue models and 27 *C. elegans* cell type models. Thus, StanDep provides a novel approach to obtain more complete context-specific models of metabolism from transcriptomics data.

## Results

### Preprocessing transcriptomics data using StanDep

Established data integration methods have struggled to consistently capture housekeeping features [13], likely because the method rely on only one or few thresholds to interpret thousands of metabolic genes. We hypothesized that interpreting similarly expressed genes together would improve the thresholding process. Accordingly, we developed a novel thresholding method that involves two steps: (1) cluster distribution of individual gene expression, considering multimeric relationships, (2) calculate and apply thresholds for each cluster of similarly expressed genes to identify a core reaction set (see Methods for additional detail). These steps (Fig 1, purple) work in tandem with a variety of model-extracting methods (Fig 1, yellow), and fit compatibly into the existing general workflow for constructing context-specific metabolic models (Fig 1, grey).

An important parameter in evaluating hierarchical clustering is the number of clusters (N). Here, the number of clusters were selected in two steps. First, we calculated the Jaccard similarity between core reaction list between any pairs of N and N+1, and then, we chose N after which Jaccard similarity between core reaction lists is over 90% (S1 Fig). As we increase the number of clusters, the weaker clusters break into smaller clusters while stronger clusters will remain. Thus, by increasing to sufficiently large number of clusters, we stabilize the selection of core reaction lists.

### StanDep seeds core reaction lists with housekeeping reactions

Currently, the common approach to identifying core reactions is to define a single 'global' threshold on gene expression [28]. All genes that are expressed above this global threshold are considered to be metabolically active, and their associated reactions constitute the core reactions. Conversely, genes that are expressed below the threshold are unconditionally interpreted as inactive. Alternatively, a recently proposed thresholding method seeks to define 'local' thresholds tailored to each gene, which is derived from the gene's average expression level across tissues [21,29]. Exceptionally high or low expression may still be unconditionally interpreted as active or inactive, respectively.

To compare these thresholding methods to StanDep, we applied the methods to a comprehensive transcriptomic profile of human cancer cell lines [30]. From these gene expression data, core reaction sets were calculated using StanDep and the following three different thresholding methods: (1) Global 75th: genes expressed above the top 25th percentile are considered active, (2) localT2: local thresholds are derived from cross-tissue mean expression; genes expressed above the top 25th percentile and below the bottom 25th percentile are interpreted unconditionally active and inactive, respectively, and (3) localT1: threshold settings are similar to localT2, but genes are never considered unconditionally inactive, even if they are below the bottom 25th percentile.

Core reactions resulting from these thresholding methods were compared to 929 housekeeping reactions associated with metabolic housekeeping genes [22] in Recon 2.2 [31]. We found that StanDep resulted in the core reaction lists with the largest fraction of metabolism-related housekeeping reactions retained (Table 1, S2 Fig).

We then analyzed the clustering captured housekeeping reactions, and how such captured reactions differed between StanDep and the other methods. Housekeeping reactions were enriched in 6 clusters; out of which, clusters 1, 2, and 11 had the largest number of housekeeping reactions (S3 Fig, Fig 2A). We found that StanDep selected genes from clusters with moderate expression (10–100 FPKM), such as cluster 1 and low expression (1–10 FPKM), such as clusters 2 and 11 (Fig 2C and 2D, S5 Fig). Indeed, we also found that housekeeping reactions that were captured by StanDep but not by the localT2 approach mostly belonged to clusters 2 and 11 (Fig 2B). By contrast, the global approach favored reactions in clusters with fewer housekeeping reactions and reactions with higher expression (>100 FPKM) (Clusters 9 and 10; S3 and S5 Figs). Altogether, the results suggest that StanDep seeds core reaction lists with housekeeping reactions that are not captured by existing methods.

## StanDep-derived core reaction lists are more self-consistent than localT2

Using StanDep, we built hundreds of models of the NCI-60 cell lines by varying 4 model uptake/secretion constraint types [32] and 6 model extraction methods (MEMs) [14–19]. The resulting models were strongly influenced by MEM used (S6A Fig), but not by constraint type (S7A Fig). As shown in Table 1, the largest number of housekeeping reactions were captured by localT2 and StanDep. Therefore, we compared StanDep models with localT2 models. Further, given that constraint types did not have strong influence over the model content, we decided to compare only the models that that were built using the exometabolomic constraints.

StanDep-based core reaction lists were larger than those from other thresholding methods (S8 Fig). We wondered if StanDep provided better support for inclusion of reactions by MEMs; thus, making them more self-consistent and independent from the extraction methods. We compared the overlap between models and their respective core reaction lists. Indeed,

**Table 1. Core reaction lists from StanDep contain highest fraction of housekeeping reactions compared to those from existing thresholding methods.**

| Thresholding method | Mean fraction of housekeeping reactions in core reaction lists across 44 NCI-60 cancer cell lines |
|---|---|
| Global 75th (global) [28] | 0.57 |
| Local T2 (25th, 75th) (localT2) [21,29] | 0.70 |
| Local T1 25th (localT1) | 0.44 |
| StanDep | 0.80 |

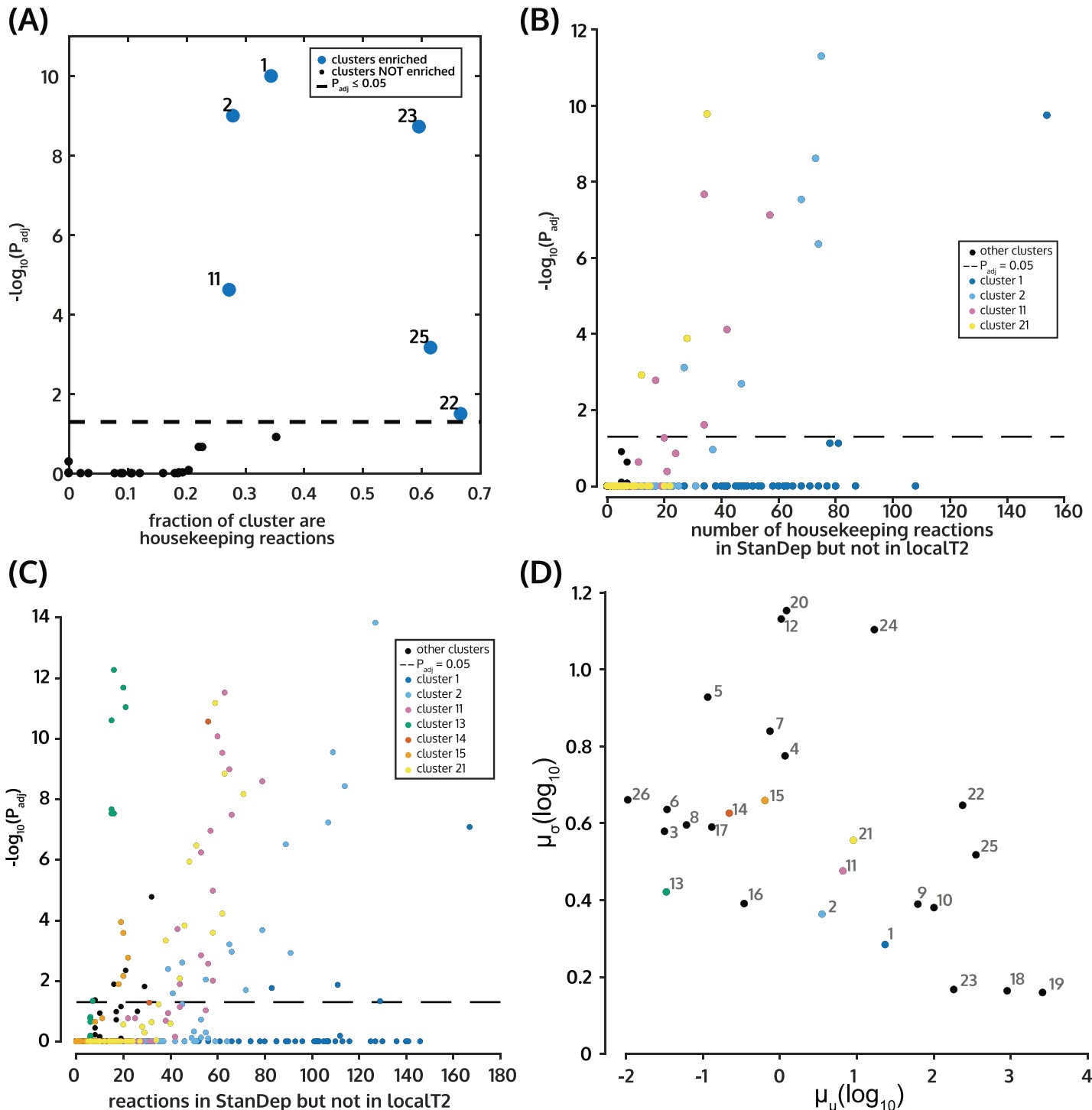

**Fig 2. StanDep enriches core reaction lists with housekeeping (HK) reactions including those lowly expressed.** (A) Housekeeping reactions are enriched in clusters 1, 2, 11, 22, 23, and 25 (big blue dots). (B) Housekeeping reactions which were captured only by StanDep (not in localT2) core reaction lists mainly belong to clusters 2 and 11. (C) Reactions which are captured by only StanDep mainly belong to clusters 1, 2, 11, and 21. (D) Standard deviation and means for each cluster. Clusters 2, 11, 13, 14, 15 and 21 which contain reaction only captured by StanDep have low mean expression and low mean standard deviation. Therefore, the reactions in these clusters are low but tightly expressed. The clusters which distinguish StanDep and LocalT2 are shown with colored dots. The colors are the same as that used in (B) and (C). Hypergeometric p-values were used for plots in A-C.

we found that for most MEMs, except MBA and GIMME, StanDep produced core reaction lists that were more self-consistent (i.e. MEMs add fewer unsupported reactions) than localT2 (S9 Fig). Thus, models built using StanDep-derived core reactions had fewer unsupported reactions compared to localT2. Further, we found that reactions that were supported by only the extraction methods but not by localT2 belonged to low expression clusters such as clusters 2 and 11 (Fig 3E). These are the same clusters that were differentially captured by StanDep (Fig 2C). Thus, these results indicate that self-consistency of StanDep should be interpreted as its ability to provide transcriptomic support for low expression reactions that were otherwise added by the MEMs.

## StanDep models accurately capture housekeeping functions

We have shown, so far, that StanDep-derived core reaction lists (S2 Fig) contained significantly more housekeeping reactions than localT2. To see if this is true for the models as well, we compared the coverage of housekeeping genes and housekeeping reactions between StanDep and localT2 models. Overall, we found that StanDep contained more housekeeping reactions than localT2 models (Fig 3A and 3B). We then wondered if the housekeeping reactions which are differentially present in StanDep models belong to specific pathways. The housekeeping reactions which were differentially present in the StanDep-derived core reaction list (S10 Fig) and models (S11 and S12 Figs) belonged to glycan synthesis and metabolism, metabolism of cofactors and vitamins, and fatty acid oxidation.

Biotin metabolism is part of the metabolism of cofactors and vitamins. We found that the housekeeping reactions belonging to biotin metabolism were differentially present in the StanDep-derived models compared to global or localT2 regardless of extraction method (S13 Fig). There are two genes part of this pathway: HLCS (reaction ID: BTNPL, Gene ID: 3141) and biotinidase (reaction ID: BTND1, Gene ID: 686 (BTD)) (Fig 3C). Among these BTND1 was identified as the housekeeping reaction. Absence of biotinidase has been identified as an inherited disorder [33] and a lifelong treatment is required. BTND1 is responsible for recycling biotin from biocytin; and low expression has also been identified as a marker for breast cancer [34,35] and thyroid cancer [36]. Out of 4 breast cancer cell lines, global-derived models were least likely to include BTND1 in breast cancer cell lines, followed by localT2-derived breast cancer cell lines. However, StanDep predicted them in over 90% of the models, regardless of extraction method.

In mammalian cells, glycosylation performs essential functions such as protein folding, targeting, stabilization, and adhesion [37]. Further, changes in glycosylation have also been reported to contribute to cancer cell physiology [38,39]. Phosphatidylinositol phosphate metabolism, part of glycan biosynthesis and metabolism, involves glycosylphosphatidylinositol (GPI)-anchor proteins which form into a complex and serves to anchor proteins to the cell surface. An example of such a complex is H8/H7/M4B transamidase (reactions IDs: H7-TAer, H8TAer, and M4BTAer; Gene IDs: 94005 (PIGS), 51604 (PIGT), 128869 (PIGU), 8733 (GPAA1), and 10026 (PIGK)). PIGT and its gene products have been hypothesized to play a role in growth of breast cancer via paxillin phosphorylation [40]. Further, overexpression of PIGT protein has also been observed in several other cancer types [41]. As with BTND1, reactions associated with PIGT had a higher coverage in StanDep-derived models than localT2- or global-derived models (S14 Fig), except in fastCORE (S15 Fig). Further, for global and localT2-derived models, housekeeping reactions from this pathway were included because of the extraction methods (S16 Fig) rather than the core reaction lists.

Given this successful application of StanDep in capturing housekeeping genes and reactions in cancer cell lines, we also applied it to transcriptomics data from Human Protein Atlas

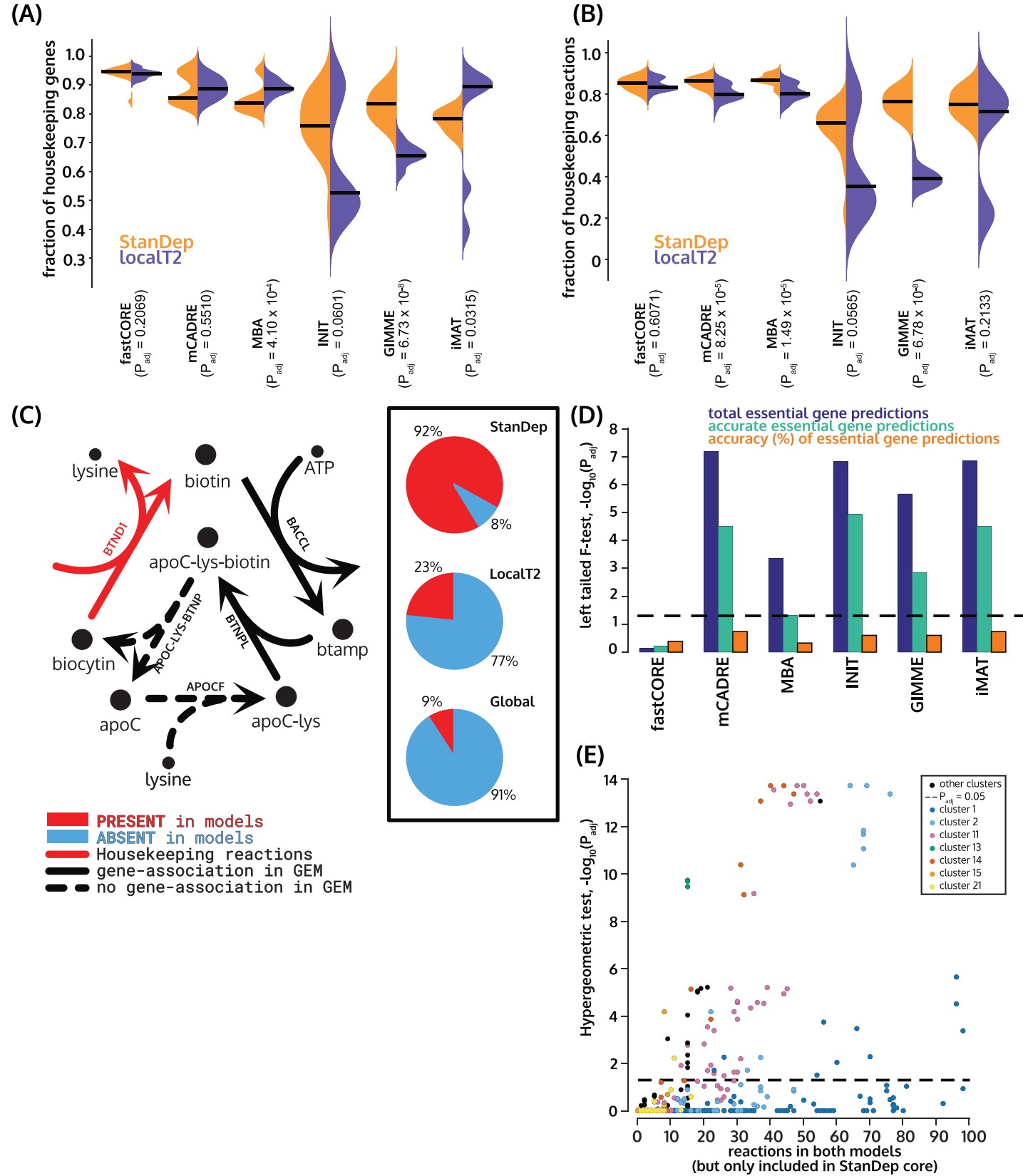

**Fig 3. Performance of StanDep models (purple) compared to localT2 models (orange) using a list of housekeeping (HK) genes/reactions, CRISPR-Cas9 essentiality screens.** Coverage of (A) housekeeping genes and (B) housekeeping reactions in StanDep and LocalT2 are shown. BHFDR correction for Wilcoxon rank-sum test p-values are reported on the x-axis. The housekeeping genes and reactions present in flux-consistent Recon 2.2 are 317 genes and 929 reactions respectively. These numbers were used for normalizing (A) and (B) respectively. (C) Biotinidase (shown in red arrows, BTND1), a housekeeping reaction in Biotin metabolism, is differentially present in the StanDep-derived models. StanDep-derived models have improved coverage of this reaction compared to localT2- or global-derived models (pie plots in inset), regardless of extraction method (S13 Fig). Housekeeping reactions are given in red, gene-associated reactions are in black, and reactions with no gene associated are given in dashed arrows. The pie plots show percentage of models (cell line-extraction method = 120 models in each pie plot) that contained (red) or did not contain (blue) the reaction. (D) StanDep models are more reliable than localT2 models for predicting gene essentiality. BHFDR correction for left-tailed F-test (alternative hypothesis: variance in total and accurate number of essential gene predictions for StanDep models is less than that of localT2 models) for comparison of gene essentiality predictions using CRISPR-Cas9 essentiality screens are plotted. (E) StanDep provides transcriptomic explanation (besides network connectivity evidence) for inclusion of lowly expressed reactions by MEMs. Each point on the plot represents the p-value of reactions belonging to a given cluster which were part of both StanDep and localT2 models but were supported by only StanDep core reaction lists.

(HPA) data [6]. We found that, across 32 human tissues in HPA, the models of human tissues extracted using StanDep resulted in 92% of housekeeping genes in fastCORE and 88% of housekeeping genes in mCADRE. The models of human tissues have been discussed in Supplementary Text (S1 Text). Thus, StanDep can be easily extrapolated across datasets.

## StanDep-extracted models accurately capture essential genes

In addition, we analyzed the content of the extracted models and evaluated their capacity to predict gene essentiality, as experimentally measured by CRISPR-Cas9-mediated loss-of-function screens. [24,25,27]. In these predictions, StanDep-derived models performed comparably to models from other methods. However, prediction accuracies from StanDep had lower variance (Fig 3D; S17 Fig), suggesting that StanDep-derived models may be more reliable in such predictions.

Lastly, we compared the list of essential genes in the two sets of fastCORE models, and then analyzed accurate predictions of matches and mismatches among the two thresholding methods. Among accurate matching predictions, we found that 113 genes were accurately predicted in at least one cell line by both models. Among these, 22.12% were ubiquitously essential while 26.54% were cell line-specific (black, S18 Fig). Among 58 genes that were accurately predicted by StanDep-derived models only, 22.41% were essential in at least ten cell lines, while 39.66% were essential in exactly one cell line (blue dotted, S18 Fig). Among these was also 6 subunits of cytochrome C oxidase, an enzyme belonging to oxidative phosphorylation. Recently, oxidative phosphorylation has garnered interest for targeting cancer due to its important role in dissipation of excess electrons [42]. While the reaction itself was present in both sets of models, only StanDep-derived models accurately identified them as essential. Interestingly, among 46 genes that were accurately predicted by localT2-derived models only, none were essential in more than seven cell lines while 58.7% were essential in only one cell line (orange dash, S18 Fig). These results suggest that StanDep-derived models contain both ubiquitously essential genes (22.12%) and cell line-specific essential genes (26.54%). Both are important determinant of quality of models.

We also extended gene essentiality analysis for models of 27 *C. elegans* cell types using previously published animal-level RNAi screens [26], transcriptomics data [7] and metabolic model [43]. However, due to the lack of systemically identified housekeeping genes for *C. elegans* or cell type-specific essential genes, we could compare only the genes which ubiquitously essential. Here, too, we found that StanDep models contained information not only about animal-level essential genes but also cell type-specific pathways were enriched in respective tissues; for e.g. peroxisomal fatty acid beta oxidation in neurons, hypodermis and intestine. The details from the models of *C. elegans* cell types are discussed in the Supplementary Text (S1 Text). These results suggest that StanDep-derived models are not only able to predict ubiquitously essential genes but also cell line-specific essential genes.

## Discussion

Several methods exist for building context-specific models by integrating transcriptomic data into genome scale models [14–19]. Previous work identified thresholding as the most influential parameter impacting the resulting model content and quality [13,21]. Despite this, established thresholding methods have struggled to reflect biological complexity. Here, we present StanDep, a novel heuristic approach for determining thresholds. We made hundreds of models using StanDep and evaluated them against models constructed using existing thresholding methods [29].

Existing thresholding methods assume that all genes have the same expression patterns, regulation, and stabilities. This, however, is not true; in particular, the metabolic products of some genes are needed in lower quantities or the proteins may be more stable, so the necessary mRNA levels may be low; e.g. glycosylation related genes have low transcript abundance [44]. A thresholding method dependent on a single arbitrary expression threshold may therefore exclude such genes. One example of such class of genes are lowly expressed housekeeping genes, consistently expressed across all cellular contexts as their gene products are required for cellular maintenance or the production of essential enzyme prosthetic groups. Indeed, classification of housekeeping genes is an important part of this study. In the literature, we found only one study that explicitly provided a list of housekeeping genes [22]. Recently a novel tool, GeneGini, has been shown to be an effective way of identifying housekeeping genes [45,46]. Housekeeping genes and reactions identified using this tool (S19 Fig) reproduced the results presented in this study (S20 and S21 Figs).

We were able to show that StanDep core reaction lists (S10 Fig) and models (S11 and S12 Figs) better captured housekeeping reactions in processes such as glycan and cofactor metabolism. Our study identified cluster 2 and cluster 11 (mean expression less than 10 FPKM, S5A Fig; standard deviation less than 2 FPKM, S5B Fig) as containing enzymes catalyzing over 300 housekeeping reactions (S3 Fig). Indeed, the success of StanDep in capturing more housekeeping reactions is primarily attributed to considering: (i) patterns of variability in gene expression, and (ii) standard deviation as a measure of biological variability in the formulation of cluster-specific thresholds. Thus, we showed that gene-specific variability is an important determinant when preprocessing transcriptomics data.

Apart from the MEMs, there can be two potential reasons for reactions to be included in the extracted model: (i) presence of reactions in core reaction list calculated using a thresholding method, or (ii) demanded by the exometabolomic constraints applied. In cases such as that of biotin metabolism for which exometabolomic constraints were not available, the reactions had to come from the selection by StanDep. Further, as shown for StanDep, increased inclusion of reactions due to core reaction list will result into a method that is more self-consistent. In cases such as that of phosphatidylinositol phosphate metabolism, the inclusion was attributed to the MEMs in localT2 and global models; and to core reaction list in StanDep models (S16 Fig). It serves as an example for reactions which make StanDep core reaction lists more self-consistent. Thus, MEMs are not entirely reliable in recapitulating critical cellular functions [13], highlighting the importance of accurately identifying core reactions during data integration.

Housekeeping reactions of both these pathways showed high coverage in not only MBA-like methods but also GIMME-like methods [47] (S13 and S15 Figs). Inclusion of such reactions also explains why the Jaccard similarity for StanDep-derived models is higher across extraction methods than localT2- or global-derived models (S22 Fig). Thus, inclusion of such housekeeping reactions is an important criterion that the models of human cells must satisfy as they not only capture the biology of human cells but also provide better agreement

regardless of extraction method. Thus, it is an important observation that the StanDep-derived core reaction lists captured >100–200 additional housekeeping reactions per cell line compared to those derived by global or localT2 approaches. Using global thresholding of top 25<sup>th</sup> percentile, the selection was favored from clusters where enzyme expression values (i.e. combined expression of enzyme-coding genes involved in a reaction) were higher (S4 Fig). However, without direct experimental evidence (which would be context dependent) there is no way to determine an exact threshold above which a gene and the associated enzyme can be classified as active. StanDep avoids using a single threshold by calculating and applying cluster-specific thresholds. Of course, our rationale for using cluster-specific thresholds was that some clusters are enriched in housekeeping reactions. Nevertheless, applying cluster-specific thresholds was possible because of diversity of the gene expression profiles across cancer cell lines and human tissues used in this study and availability of datasets with larger samples. However, since StanDep requires larger datasets with higher diversity of gene expression, a global approach may still be an appropriate choice when integrating highly homogenous or small transcriptomics data sets.

We also benchmarked StanDep using six of the existing MEMs to build models for the NCI-60 cancer cell lines. This showed that StanDep works best with MBA-like extraction methods. Besides using housekeeping genes/reactions and CRISPR-Cas9 gene essentiality screens for validation and comparison with localT2-derived models. In light of StanDep-derived models being at least comparable to localT2-derived models in accuracy (Fig 3D), we can say that StanDep provided a transcriptomic explanation for why a reaction needs to be included within the model making self-consistency an important quality metric.

Context-specificity of a model can be shown in two ways: (i) capturing reactions that do occur in a context, and (ii) not capturing reactions that do not occur in a context. In this paper, we have used the former way. However, it is difficult to qualify a reaction to be a false positive reaction for a model. For example the peroxisomal fatty acid beta-oxidation pathway in *C. elegans* was enriched in hypodermis, intestine, and rectum. While we were able to find studies showing the role of this pathway in hypodermis and intestine, we did not find any for rectum. This could be because the cell type hasn't garnered enough interest within the *C. elegans* community. Therefore, it is difficult to argue in such models if presence of a reaction is false-positive or novel discovery.

The level of diversity in gene expression across cancer cell lines and human tissues helped in the identification of housekeeping genes. The models of cancer cell lines and *C. elegans* cell types were also able to predict essential genes. Furthermore, we showed that StanDep can capture relevant aspects of cell type-specific metabolism, such as the presence of peroxisomal fatty acid $\beta$-oxidation in neurons, intestine, and hypodermis cell types of *C. elegans* and cell line-specific essential genes of NCI-60 cancer cell lines. In conclusion, StanDep demonstrates that in addition to considering the expression level of a gene, the use of its variability across tissues and cell types can help to better define context-specific cellular function.

## Methods

### Datasets used

For this study, we used gene expression data for NCI60 cancer cell lines [30], HPA gene expression data for tissues [6], and gene expression data for *C. elegans* cell types [7]. For validating our models, we used a list of housekeeping genes [22], CRISPR data for 20 NCI60 cancer cell lines [24,25,27], and RNAi phenotypic data for *C. elegans* [26]. For further details on data extraction, please see supplementary methods (S1 Text).

## Data processing

We selected genes from the human metabolic reconstruction, Recon 2.2 [31]. This included for NCI60 data, 1416 genes; for HPA data, 1661 genes out of 1673 genes in Recon 2.2; and for *C. elegans* cell type data, 1248 genes were part of the global expression datasets and *C. elegans* GEM [43]. We then converted gene expression values into enzyme expression values using gene mapping. We note that we do not mean "abundance of enzyme". Here on we use the term "enzyme expression" to describe combined expression of enzyme-coding genes for a reaction. Gene mapping involved extracting gene-protein-reaction (GPR) relationships from the model and calculating enzyme expression. The extraction of GPR was done using the COBRA function, *GPRparser.m*. For enzymes that have only one subunit, the value of enzyme expression is same as the value of gene expression. For multimeric enzymes, these relationships share an "AND" relationship; thus, the minimum value amongst genes part of the enzymes were set as the enzyme expression value. The assumption for multimeric enzymes was that gene with lowest expression will govern the amount of functional enzyme expressed. It should be noted that enzyme complex stoichiometry was not considered as it is not fully described for all the enzymes involved in the network reconstructions used here [48]. We did not resolve OR relationships representing isoenzymes and allowed all functional enzymes to be represented in the enzyme expression dataset. The enzyme expression data spanned 1325 enzymes (4133 reactions) for NCI60 data, 1792 enzymes for HPA data, and 2533 enzymes for *C. elegans* data.

## Hierarchical Clustering

**Clustering distribution patterns of gene expression.**  We $\log_{10}$-transformed the calculated enzyme expression dataset and counted the number of samples expressed with each bin width. Bin width were set based on the $\log_{10}$-transformed minimum and maximum enzyme expression values. This resulted in a matrix with rows representing each enzyme, columns representing bins, the value within the matrix representing number of samples from the dataset which were expressed within each bin range. We then performed hierarchical clustering with *Euclidean* distance metric and *complete* linkage metric to cluster genes based on distribution pattern of gene expression. We also show the comparison between using other distance (S23 Fig) and linkage methods (S24 Fig, S1 Text).

**Deciding number of clusters.**  Clustering in our work is used as a tool to divide genes into categories based on distribution patterns of their expression across different conditions. These clusters are then responsible for generating their own threshold. The number of clusters were chosen such that any further increase in the cluster results in large similarity (>90%) in core reaction lists. Therefore, by setting the number of clusters sufficiently large, we stabilize the selection of core reactions. For the NCI60 Klijn et al. dataset, we used 26 clusters (S34 Fig); for HPA dataset, we used 19 clusters; and for *C. elegans*, we used 14 clusters for enzyme expression and gene expression data respectively. We also show the comparison of choosing different number of clusters (S1 Fig; S1 Text Supplementary Results).

**Clustering core reaction sets or models.**  For analysis of models, we calculated Jaccard similarity of reaction content across different models which were part of any given analysis. We then performed hierarchical clustering to see how tissues are grouped. Hierarchical clustering was performed with the *Euclidean* distance metric and *complete* linkage metric. The interpretation of clustering Jaccard similarity is that models that are most similar to each other are likely to be equally far from other models.

## Identification of Core Reactions

**StanDep.**  StanDep applies thresholds specific to each cluster of genes (Fig 1). In the StanDep threshold formulation, we included two terms: (i) standard deviation, and (ii) mean term.

Fine-tuned expression level of genes is represented as the Standard deviation term; and is dependent on the difference between standard deviation of the cluster and the dataset. Lower standard deviation favors the selection of enzymes in all contexts while higher standard deviation term reflects context-specificity of the enzymes. The mean term, interpretation of second assumption, is dependent on the magnitude of the expression of enzymes in that cluster. In both cases, we used the difference between cluster and overall data to address inconsequential variations that maybe occurring in expression. The standard deviation is always positive but logarithmic mean may be negative and sometime be even quite large. Therefore, we introduced normalization to make the standard deviation term and mean term at par. The threshold for each cluster is given by the following equations:

$$\Theta_c = (\theta_c - min(\theta_c)) * 100/max(\theta_c - min(\theta_c)); \;\; \Theta_c \in [0, 100] \tag{1}$$

$$\theta_c = f(\sigma_c) + g(\mu_c); \tag{2}$$

$$f(\sigma_c) = (\sigma_c - \Delta)/max(\sigma_c - \Delta); \tag{3}$$

$$g(\mu_c) = -(\mu_c - \boldsymbol{M}); \tag{4}$$

In the above set of equations, $\boldsymbol{\Theta_c}$ is the processed threshold value for a given cluster $c$; $\boldsymbol{\theta_c}$ is the raw value of threshold for cluster $c$; $\boldsymbol{\sigma_c}$ is the standard deviation of the cluster $c$; $\boldsymbol{\Delta}$ is the standard deviation of the dataset; $\boldsymbol{\mu_c}$ is the mean of the cluster $c$; and $\boldsymbol{M}$ is the mean of the dataset. The equation is derived by penalizing cluster-specific thresholds based on: (i) how low the cluster mean is compared to the mean of the dataset; (ii) how far the standard deviation of the cluster is from the standard deviation of the dataset. The final normalization was done to ensure that the clusters-specific thresholds are between 0 and 100. The $\boldsymbol{\Theta}$ is the top percentile value of the cluster-specific data above which an enzyme in that cluster in a given context is qualified active. If the value of $\boldsymbol{\Theta_c}$ is 100, we set the threshold value of the cluster as the mean of the data.

The current published literature on the below thresholding methods does not address how the threshold values should be derived. Therefore, we used some of the most commonly used percentile values in previously published studies [13,21,28,29].

**Global thresholding.** For global thresholding scheme, we used top 25th percentile of the global experimental dataset. The method has been used in previous studies [13,21,28]. Enzymes that have an expression value above the threshold were considered to active. The reactions catalyzed by these active enzymes were identified as core reactions.

**LocalT1 thresholding.** For the localT1 thresholding approach, we used only one value; top 75th percentile. The method has been used in previous studies [21]. Enzymes that have the mean (across all contexts in the dataset) expression value below the threshold were considered as inactive in all contexts. All other enzymes were threshold at mean of the enzyme and were considered as active if the enzyme expression was above the mean of the enzyme expression. The reactions catalyzed by these active enzymes were identified as core reactions.

**LocalT2 thresholds.** For localT2 thresholding, we used top 25th (upper threshold) percentile and top 75th percentile (lower threshold). The method has been used in previous studies [21,29]. Enzymes that have the mean (across all samples in the dataset) expression value below the lower threshold were considered as inactive in all contexts. Further, enzymes that have the mean (across all samples in the dataset) expression value above the upper threshold were considered as active in all contexts. The remaining enzymes were provided thresholds at the mean of the enzyme expression and were considered as active if the enzyme expression

was above the mean of the enzyme expression. The reactions catalyzed by these active enzymes were identified as core reactions.

## Constraining Pre-extraction Models and Model reduction

**Exometabolomic constraints.** Exometabolomic data of the NCI60 cell line were obtained from previous work [32] and further processed as previously described [13]. After processing, we added 23 new demand reactions, wherein each reaction is secreting a different metabolite. These were added to reflect the experimental observations by Jain et al. The biomass reaction was changed to one that contains precursor molecules from the one that contains DNA, RNA, protein, lipids, carbohydrate, and small molecules. The average composition of these components were determined from literature [49–59]. The stoichiometric coefficients of the metabolites in the biomass equation are presented in S1 Table. Please see Methods of Opdam et al. 2017 for how these stoichiometric coefficients were calculated [13]. The replacement of the biomass reaction was done to all the models as discussed in detail previously [13,29]. The global lower and upper bounds for all reactions except biomass and ATP demand were set to -1000 and 1000 respectively. The lower bounds of the biomass reaction and ATP demand were constrained to relatively small values of the order of 1e-2 and 1.833 mmol gDW$^{-1}$ h$^{-1}$ [60] respectively. The cell line specific constraints on 78 demand and exchange reactions were applied on the modified Recon 2.2, followed by making flux consistent constrained genome-scale models for each of the cell lines. This was done by identifying and removing flux-inconsistent reactions using *fastcc.m* in COBRA Toolbox. The flux tolerance was always set to 1e-8.

**No constraints.** To make unconstrained models, we did not apply exometabolomic constraints but only applied constraints on lower bounds of biomass and ATP demand reaction as described above. The global lower and upper bounds were set to -1000 and 1000 respectively. This was followed by identifying and removing flux inconsistent reactions. The flux tolerance was always set to 1e-8.

**Semi constrained.** To make semi-constrained models, we applied directional constraints on demand and exchange reactions of each cell line, applied constraints on lower bounds of biomass and ATP demand as described above. The global lower and upper bounds were set to -1000 and 1000 respectively. This was followed by identifying and removing flux-inconsistent reactions. The flux tolerance was always set to 1e-8.

**Relaxed constraints.** To make relaxed models, we constrained the direction of flow to 10 mmol gDW$^{-1}$ h$^{-1}$ on demand and exchange reactions as suggested by exometabolomic data. The order of magnitude of original constraints on these reactions was 1e-3 to 1e-6. The global lower and upper bounds were set to -1000 and 1000 respectively. This was followed by identifying and removing flux-inconsistent reactions. The flux tolerance was always set to 1e-8.

## Implementation with model extraction methods (MEMs)

In this study, we compared the models derived using localT2 and StanDep. This section describes the extraction of StanDep-derived models by tailoring each of the MEMs. Models derived using localT2 were not constructed in this study, rather we extracted those models from a previous study [29]. Therefore, for implementation of each of the extraction methods with these thresholding methods, please see the methods for that study.

To construct models using 6 of the extraction methods these inputs were common to all: (i) a flux-consistent Recon 2.2 genome-scale model was used, and (ii) epsilon, a.k.a. flux tolerance, was set to 1e-8. Inputs specific to a given MEM are described below.

**FASTCORE.** To construct models using FASTCORE [19], we used *fastcore.m* in the COBRA toolbox. Other inputs needed for the algorithm are requires core reaction lists. Please

see above on how we identified them. The biomass reaction was manually added to the core reaction list.

**iMAT.** To construct models using iMAT [18], we used *iMAT.m* in the COBRA toolbox. Other inputs needed for the algorithm are: (i) core reactions (i.e., list of reactions identified to be active, including the biomass reaction) and (ii) non-core reactions, which are not part of core reactions (reactions not associated to a gene were not included in non-core reactions).

**MBA.** To construct models using MBA [14], we used *MBA.m* in the COBRA toolbox. Other inputs needed for the algorithm are: (i) high expression set, list of reactions which are highly expressed and (ii) medium expression set, list of reactions which are moderately expressed. We generated 10% interval around threshold for each cluster. We defined high expression set as the list of reactions catalyzed by enzymes which are above 110% of the threshold value, and medium expression set as the list of reactions catalyzed by enzymes which are between 90% and 110% of the threshold value. For instances where a reaction was present in both high and medium expression set, we interpreted it as at least enzyme associated to the reaction being able to express at high levels. Thus, we put these reactions in high expression sets. The biomass reaction was given the highest value.

**mCADRE.** To construct models using mCADRE [15], we used *mCADRE.m* in the COBRA toolbox. Other inputs needed for the algorithm are: (i) ubiquity score (i.e., how often a reaction is expressed across samples of the same context); (ii) confidence scores quantifying level of evidence for a reaction to be present in the model; (iii) protected reactions; and (iv) since we did not protect any reactions, we set the functionality check to 0. To calculate ubiquity score ($U_{c,i}$), we calculated threshold distances ($D_{c,i}$), here defined as distance of a given enzyme expression ($x_{i,c}$) in the context $i$ from the threshold ($\Theta_c$) of the cluster $c$ where the enzyme belongs. The threshold distances and ubiquity scores were calculated using the Eqs (5–7).

$$D_{c,i} = x_{i,c} - \Theta_c \tag{5}$$

$$if \ D_{c,i} > 0; \ U_{c,i} = 1 \tag{6}$$

$$if \ D_{c,i} < 0; \ U_{c,i} = 1 - (D_{c,i}/min(D_{c,i})) \tag{7}$$

We used the ubiquity score to quantify how often an enzyme is expressed in samples of the same context. For isoenzymatic reactions, the reaction ubiquity score was set to the enzyme with maximum ubiquity score. For reactions which do not have an associated gene, the ubiquity score was set to -1. Since, we did not have confidence scores, we assigned a confidence of 0 to all reactions, as suggested in COBRA toolbox tutorial for mCADRE. However, we also tried using our list of core reactions as a binary vector specifying whether a reaction is in the core set and if it did not have any effect of the final model. The biomass reaction was manually assigned a ubiquity score of 1. The confidence score of 1 is associated with transcriptomics evidence and our metric ubiquity score already has this information.

**INIT.** To construct models using INIT [16], we used *INIT.m* in the COBRA toolbox. Other inputs needed for the algorithm are reaction weights, varying between -1 and 1. To calculate enzyme weights, we calculated the threshold distance for each enzyme as described previously, without normalizing. Weights for all reactions catalyzed by an enzyme were same as the enzyme weight. Here, we used a different normalizing scheme. We scaled our threshold distances to a maximum threshold distance for any of the enzymes within the data. For isoenzymatic reactions, the weights of each enzyme were added. We set the weights for non-gene associated reactions to 0. The biomass reaction was manually assigned a weight of 1.

**GIMME.** To construct models using GIMME [17], used *GIMME.m* in the COBRA tool-box. Other inputs needed for the algorithm are: (i) a reaction expression vector representing gene expression values associated with the reactions; and (ii) a threshold determining whether reaction expression is considered active. We calculated the reaction expression vector in the same way as we calculated enzyme weights for INIT. The thresholds were set to 1. The biomass reaction was given a value of 1.

### Gene essentiality in NCI60

**NCI60 data.** To test our essentiality predictions of NCI60 models with CRISPR screen data, we downloaded pooled CRISPR knockout screen data from DepMap.org [24,25,27] for 20 NCI-60 cell lines. Essential genes were identified based on the CRISPR score. The CRISPR score was calculated as the ratio of abundance of single guide RNA (sgRNA) of a knock out after and before growth selection. A negative CRISPR score suggests a higher probability that the gene is essential. The accuracy was estimated using the percentage of predicted essential genes that have a negative score [61]. We then used 1-tailed Wilcoxan rank sum test to identify if the CRISPR scores for genes predicted to be essential in the metabolic model and CRISPR scores of genes predicted to be non-essential are coming from the same populations.

**RNAi phenotypic data.** To get the list of essential genes in *C. elegans*, we extracted genes that presented a *Nonv* or *Gro* RNAi phenotype. As described by the authors [26], *Nonv* pheno-type refers to all phenotypic classes that result in lethality or sterility (1170 essential genes); and *Gro* refers to phenotypic classes that result in growth defects, slow post-embryonic growth or larval arrest (276 essential genes). Out of these, the iCEL1273 [43] model contained 187 genes. Similarly, we found 900 non-essential genes in iCEL1273.

## Supporting information

**S1 Text. Supporting text containing supplementary methods and results.**
(DOCX)

**S1 Table. The stoichiometric coefficients of the metabolites involved in the biomass equation.**
(XLSX)

**S2 Table. The metabolic task reports for models of human tissues extracted using StanDep-fastCORE, StanDep-mCADRE, global-fastCORE, and global-mCADRE.**
(XLSX)

**S1 Fig. Comparison of Jaccard similarity for core reaction list of 44 cancer cell lines using (B) different number of clusters and (A) silhouette value for quality of clusters when using StanDep.** The *complete* linkage method and *Euclidean* distance were used. More than 10 clusters lead to over 90% mean Jaccard similarity.
(JPG)

**S2 Fig. StanDep core reaction lists contain highest number of housekeeping reactions across all cell lines.**
(JPG)

**S3 Fig. Clusters are enriched in housekeeping reactions. Enriched clusters are shown in blue bars (left axis).** Clusters containing housekeeping reactions but are not enriched are shown in black bars (left axis). The fraction of reactions in each cluster that are housekeeping reactions are given in red dots (right axis). BHFDR correction was used for hypergeometric

test for over-representation.
(JPG)

**S4 Fig. Reactions captured by global approach are enriched in clusters which have higher expression.** Each dot is a cell line-cluster pair.
(JPG)

**S5 Fig.  Mean (A) and standard deviation (A) of expression of each enzyme (black dots) and that of the clusters (red dots) is shown.** The blue lines are the same for the entire data.
(JPG)

**S6 Fig. Comparison of reaction content of models built using StanDep.** (A) Distribution of Jaccard similarity between models built using different constraint types, extraction methods, and belonging to different cell lines. (B) Distribution of Jaccard similarity between models extracted using different extraction methods.
(JPG)

**S7 Fig. Comparison of reaction content of models built using StanDep.** (A) Distribution of Jaccard similarity between models built using different constraint types, extraction methods, and belonging to different cell lines. (B) Distribution of Jaccard similarity between models extracted using different extraction methods. (C) Heatmap of Jaccard similarity between models extracted using mCADRE and fastCORE. (D) Heatmap of Jaccard similarity between models extracted using MBA and fastCORE.
(JPG)

**S8 Fig.  Comparison of StanDep (A) core reaction lists and (B) models with localT2.** (A) On y-axis, mean number of core reactions across all 44 NCI-60 cancer cell lines. (B) The size of the models is determined by number of reactions in the model.
(JPG)

**S9 Fig. StanDep core reaction lists are more self-consistent than localT2.** Except GIMME, the overall dissimilarity between core reaction lists and the extracted models are at least similar (MBA) or lower (fastCORE, mCADRE, INIT, and iMAT) than the same for localT2 models.
(JPG)

**S10 Fig.**  Box plot of number of cell lines in which housekeeping reactions differentially present and enriched in StanDep-derived core reaction lists compared to that of (A) global or (B) localT2. Each dot represents one cell line where the housekeeping reactions differentially present in StanDep belonging to a pathway were statistically significant. The colors indicate number of cell lines where the pathway was significant. The p-values were corrected using BHFDR.
(JPG)

**S11 Fig.  Box plot of number of cell lines in which housekeeping reactions differentially present and enriched in StanDep-derived models compared to that of localT2 thresholding using (A) fastCORE, (B) mCADRE, (C) MBA, (D) INIT, (E) GIMME, and (F) iMAT.** Each dot represents one cell line where the housekeeping reactions differentially present in StanDep-derived models belonging to a pathway were statistically significant. All models were built using exometabolomic constraints. The colors indicate number of cell lines where the pathway was significant. The p-values were corrected using BHFDR.
(JPG)

**S12 Fig.  Box plot of number of cell lines in which housekeeping reactions differentially present and enriched in StanDep-derived models compared to that of global thresholding**

**using (A) fastCORE, (B) mCADRE, (C) MBA, (D) INIT, (E) GIMME, and (F) iMAT.** Each dot represents one cell line where the housekeeping reactions differentially present in Stan-Dep-derived models belonging to a pathway were statistically significant. All models were built using exometabolomic constraints. The colors indicate number of cell lines where the pathway was significant. The p-values were corrected using BHFDR.
(JPG)

**S13 Fig. Binary heatmap of reactions from Biotin metabolism present in models built using (A) StanDep, (B) LocalT2, (C) Global and different extraction methods for each of the 20 cancer cell lines.** Black indicates the absence of reaction in the model and color indicates presence. Highlighted in blue are breast cancer cell line models where BTND1 has been identified as a marker.
(JPG)

**S14 Fig. Reactions part of Glycosylphosphatidylinositol-anchor biosynthesis (phosphatidylinositol phosphate metabolism in Recon 2.2) and their coverage across models (insets containing pie plots) built using StanDep (left), localT2 (middle), and global (right).** The genome-scale model (GEM) used here was Recon2.2. Housekeeping reactions are given in red, gene-associated reactions are in black, and reactions with no gene associated are given in dashed arrows. The pie plots show percentage of models (cell line-extraction method = 120 models in each pie plot) that contained (red) or did not contain (blue) the reaction.
(JPG)

**S15 Fig. Binary heatmap of reactions from Phosphatidylinositol phosphate metabolism present in models built using (A) StanDep, (B) LocalT2, (C) Global and different extraction methods for each of the 20 cancer cell lines.** Black indicates the absence of reaction in the model and color indicates presence. Highlighted in blue are breast cancer cell line models where BTND1 has been identified as a marker.
(JPG)

**S16 Fig. Source of the housekeeping reactions from Phosphatidylinositol phosphate metabolism) in each of the models built using StanDep (blue), localT2 (cyan) and global (yellow) as either due to extraction method (i.e. only in the model but not in core reaction list, A) or due to core reaction list (i.e. the reaction is in the model because it was present in the core reaction list, B).** As seen, localT2 and global-derived models were likely to include due to extraction methods, yet the reactions were generally present in fewer models compared to those derived using StanDep.
(JPG)

**S17 Fig. StanDep models have higher predictability than localT2 without sacrificing accuracy of essential gene predictions.** (A) Distribution of total gene essentiality predictions generated by models. (A) Distributions of accurate gene essentiality predictions generated by models. (C) Distribution of accuracy of gene essentiality predictions. The violin plots represent distributions of StanDep (orange) and localT2 (purple) models.
(JPG)

**S18 Fig. StanDep explains a spectrum of genes which may be ubiquitous (essential to all cell lines) to cell line specific (essential to only one cell line).** The figure only contains fastCORE models.
(JPG)

**S19 Fig. Identifying a Gini coefficient (GC) cutoff from two different datasets. (A)** Normalized GC of housekeeping genes listed by Eisenberg & Levanon [22] when using HPA or Klijn et al. (NCI-60) transcriptomic data. Most of these genes have low GC value. **(B)** GC cutoff at the point where the Jaccard similarity in the housekeeping genes in both datasets intersected the highest fraction of novel housekeeping genes identified by either dataset. This value was 0.24.
(JPG)

**S20 Fig. Clusters are enriched in housekeeping reactions identified using Gini coefficient. Enriched clusters are shown in blue bars (left axis).** Clusters containing housekeeping reactions but are not enriched are shown in black bars (left axis). The fraction of reactions in each cluster that are housekeeping reactions are given in red dots (right axis). Similar to the Eisenberg-housekeeping reactions, there Cluster 1, 2, and 11 contain most of the housekeeping reactions. BHFDR correction was used for hypergeometric test for over-representation.
(JPG)

**S21 Fig. Comparison of coverage of (A) Eiseberg- and (B)GINI-housekeeping reactions in StanDep vs localT2 models from different extraction methods.** The source of housekeeping reactions does not affect the statistical significance of the model comparisons.
(JPG)

**S22 Fig. Histogram of mean Jaccard similarity of each cell lines across different extraction methods for models built using StanDep, lovalT2, and global thresholding approaches.** The figure shows higher consensus among models built using StanDep but different extraction methods.
(JPG)

**S23 Fig. Comparison of Jaccard similarity for core reaction list of 44 cancer cell lines between (B) various distance metrics and (A) silhouette value for quality of clusters for each distance metric when using StanDep.** The *complete* linkage method and *26* clusters were used. All distance metrics lead to over 90% mean Jaccard similarity.
(JPG)

**S24 Fig. Comparison of Jaccard similarity for core reaction list of 44 cancer cell lines using (B) different linkage methods and (A) silhouette value for quality of clusters when using StanDep.** The *Euclidean* distance method and *26* clusters were used. Complete, centroid, average, and median lead to nearly 90% mean Jaccard similarity.
(JPG)

**S25 Fig. Comparison of (A) reaction content and (B) gene content of models extracted using mCADRE (x-axis) and fastCORE (y-axis).** The lower inset shows the Jaccard similarity of (A) reaction or (B) gene content between these two extraction methods for each cell type. The dendrograms on left (fastCORE) and bottom (mCADRE) of the heatmaps were created using hierarchical clustering of the Jaccard similarity between content of models. The black dots on the heatmap track similar cell types across the two MEMs. The reaction content of models belonging to neuronal cell types clustered together when either MEMs were used. Models of a given cell type are best matched (black dots) with themselves across the models extracted using these two methods.
(JPG)

**S26 Fig. Validation of models extracted with fastCORE using (A) essential genes, (B) non-essential genes; with mCADRE using (C) essential genes, and (D) non-essential genes**

**obtained from RNAi screens of Kamath et al.** Accuracy of essential gene predictions is comparable to that of unconstrained models of NCI-60 cancer cell lines. Comparison with randomly permuted gene labels for presence of essential genes in the cell type models is presented in S29 Fig. Large fraction of essential genes predicted are present in all cell types.
(JPG)

**S27 Fig. Coverage and pathway classification of 187 essential genes identified by Kamath et al [26] in StanDep models of C. elegans cell types (transcriptomic data: [7] extracted using mCADRE.**
(JPG)

**S28 Fig. Comparison of mean essential/non-essential gene content for StanDep-derived models extracted using fastCORE and mCADRE with mean of 1000 random permutations of genes.** The bars represent mean percentage of 187 essential genes or 900 non-essential genes across all cell types. Error bars represent variation of the same among all cell type models. Percentage of essential and non-essential genes for models extracted using fastCORE and mCADRE is significantly different from that if models were created randomly.
(JPG)

**S29 Fig. Heatmap of fraction of (A) genes or (B) enzymes in a given cluster for a given cell line (y-axis) are binned according to their $\log_{10}$ expression value (x-axis) for Cao et al. dataset for *C. elegans*.** Heatmaps are shown for all (A) 18 clusters for gene expression and (B) 14 clusters for gene expression. Black represents all the genes in that cluster are binned in a certain expression range for a given cell type. White represents none of the genes in that cluster binned in a certain expression range for a given cell type.
(JPG)

**S30 Fig. Enrichment analysis for presence of 187 whole animal essential genes (A) extracted from RNAi screens of Kamath et al., and (B) determined from models extracted using mCADRE for clusters calculated from Cao et al. *C. elegans* cell type sciRNA-seq data.** Both analyses show enrichment in same clusters, except clusters 8 and 15 which total to only 7 genes. Clusters not enriched for whole animal essential genes are shown by black bars, while clusters enriched in whole animal essential genes are shown in blue bars. The red dots show the fraction of number of genes in each cluster are housekeeping genes (right y-axis). Enrichment was calculated as hypergeometric p-value $\leq 0.05$.
(JPG)

**S31 Fig. Heatmap of fraction of enzymes in a given cluster for a given cell line (y-axis) are binned according to their $\log_{10}$ expression value (x-axis) for Uhlen et al. (HPA) dataset for 32 human tissues.** Heatmaps are shown for all 19 clusters. Black represents all the enzymes in that cluster are binned in a certain expression range for a given cell line. White represents none of the enzymes in that cluster binned in a certain expression range for a given tissue.
(JPG)

**S32 Fig. StanDep models of human tissues (transcriptomic data: HPA [6]) extracted using (A) fastCORE and (B) mCADRE contain over 88% of housekeeping reactions.**
(JPG)

**S33 Fig. Distribution of metabolic tasks passed by models of human tissues extracted using (A) fastCORE and (B) mCADRE.** StanDep models (blue) tend to pass more tasks than global models (red).
(JPG)

**S34 Fig. Heatmap of fraction of enzymes in a given cluster for a given cell line (y-axis) are binned according to their $\log_{10}$ expression value (x-axis) for Klijn et al., 2014 dataset.** Heatmaps are shown for all 26 clusters. Black represents all the enzymes in that cluster are binned in a certain expression range for a given cell line. White represents none of the enzymes in that cluster binned in a certain expression range for a given cell line.
(JPG)

## Author Contributions

**Conceptualization:** Chintan J. Joshi.

**Data curation:** Chintan J. Joshi.

**Formal analysis:** Chintan J. Joshi, Song-Min Schinn, Anne Richelle, Isaac Shamie.

**Funding acquisition:** Eyleen J. O'Rourke, Nathan E. Lewis.

**Investigation:** Chintan J. Joshi.

**Methodology:** Chintan J. Joshi.

**Software:** Chintan J. Joshi.

**Supervision:** Nathan E. Lewis.

**Validation:** Chintan J. Joshi.

**Visualization:** Chintan J. Joshi.

**Writing – original draft:** Chintan J. Joshi, Song-Min Schinn, Anne Richelle, Isaac Shamie.

**Writing – review & editing:** Chintan J. Joshi, Song-Min Schinn, Anne Richelle, Isaac Shamie, Eyleen J. O'Rourke, Nathan E. Lewis.

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
