## [Decision Letter · Decision Letter 0]

15 Oct 2019

Dear Dr Lewis,

Thank you very much for submitting your manuscript 'StanDep: capturing transcriptomic variability improves context-specific metabolic models' for review by PLOS Computational Biology. Your manuscript has been fully evaluated by the PLOS Computational Biology editorial team and in this case also by independent peer reviewers. The reviewers appreciated the attention to an important problem, but raised some substantial concerns about the manuscript as it currently stands. While your manuscript cannot be accepted in its present form, we are willing to consider a revised version in which the issues raised by the reviewers have been adequately addressed. We cannot, of course, promise publication at that time.

Sincerely,

Anders Wallqvist

Associate Editor

PLOS Computational Biology

Daniel Beard

Deputy Editor

PLOS Computational Biology

[LINK]

Reviewer's Responses to Questions

**Comments to the Authors:**

Reviewer #1: Context-specific metabolic networks that integrate omics data with metabolic networks have been widely used to make important biological discoveries from microbes to human. One of the most important decisions in making context-specific models is defining a threshold above which a reaction is considered active. Despite the importance, the decision is often made somewhat arbitrarily and without much guidelines or consensus in the modeling field. The authors presented StanDep, a rigorous method that uses clustering to mitigate the arbitrariness in making this decision, potentially improving the robustness and consistency of context-specific models in the future. The authors did extensive benchmarking to demonstrate the performance of StanDep. StanDep is a new valuable tool for the metabolic network modeling community. However, this novel method and the manuscript itself might benefit from a few modifications suggested below.

Major:

StanDep included larger fraction of housekeeping reactions. But does it also included potentially false-positive reactions that should not be in the model?

Besides increasing the coverage of house-keeping genes/reactions, does StanDep improve models’ ability to simulate tissue-specific metabolic functions (e.g., liver metabolic tasks)?

In the StanDep-extracted models accurately capture essential genes, the authors seem to suggest that 58 genes that were accurately predicted by StanDep-derived models only are better supported by CRISPR loss of function screens. In addition to the high-level statistic, can specific examples be provided that the lethal genes uniquely captured by StanDep are biologically meaningful?

Figure 2B-D are not clearly explained. What does the numbering on x-axis mean, how are they ordered, how is the p-value on y-axis calculated? The figure legend mentioned that “(C) Reactions which are captured by only StanDep mainly belong to low but tightly expression clusters.” However, there is no expression level information in Figure 2C.

Minor:

MEM was first used on page 5, line 138 in the main text without fully spelling it out as model extraction method. Although it is spelled out in Figure 1 caption, it still helps reading if it is also described in main text as well.

Self-consistent is not quantitatively defined in page 5, line 139-142. Does higher self-consistency mean that the core reactions defined by StanDep require less non-core reactions to support?

In figure legends, HK is used as an acronym for housekeeping. To some people, HK is often an acronym for hexokinase, a well-known, classical metabolic reaction, causing potential confusion.

Reviewer #2: The paper of Joshi et al. deals with an issue arising during the generation of context-specific metabolic networks based on transcriptomics data. There are already several types of algorithms (reviewed and categorized in reference [45]) which can be used for context-specific metabolic networks, and majority of them require setting a threshold value for declaring a reaction active. This threshold is either set to hold for all genes or is gene-specific. The authors propose a heuristic, StanDep, that takes a middle road, whereby the threshold value is determined per groups of genes. The extracted models from different scenarios (i.e. conditions and organisms) are partially tested against lists of housekeeping genes, CRISPR data, and RNAi phenotypic data. Therefore, the main contribution is the way to set the threshold, from which point the study depends on existing constraint-based approaches for context-specific network extractions.

I find that this a small contribution which also suffers from additional issues, which I detail below:

1. The author claim that the thresholding method involves: (1) cluster distribution of individual gene expression, considering multimeric and isozyme relationships (line 86). Is there support that the clusters contain protein comlplexes and / or isozymes. To the best of my knowledge, co-expression / transcript similarity can partly explain protein complexes, but only if complex stoichiometry is factored in; in addition, this rarely holds for isozymes. The authors should provide evidence for this claim.

2. The first step in the threshold selection requires hierarchical clustering based on Euclidean distances between gene expression profiles. While it is obvious that magnitude of gene expression is taken into consideration by following this approach, the reasons for not using other approaches (results shown in Supplementary Figures), are not provided. The results in the Supplementary Figures must be critically discussed in the main text—for instance, why is the comparison done for 26 clusters? Do all similarity measures yield the same number of clusters when following the procedure described in the Methods?

3. The approach for selecting the number of clusters does not correspond to the steps described in the Methods. Lines 313 and 314 state that “the number of clusters is determined such that all pathway[s] is enriched in at least one cluster”. This is a—major—concern for three reasons: (1) it is difficult to imagine how this condition can be ensured, and if so, that this results in a single number of clusters, (2) the logic for selection of thresholds is then circular, since every pathway will get respective thresholds, thus partly ensuring its inclusion of the context specific model. The bottom line is that the selection of thresholds—is not—based only on the transciptomics data, as claimed, but also includes the model structure (i.e. pathway information), and (3) Jaccard distance of core reactions, mentioned on lines 91-94, is not used for selection of thresholds, but for comparison of the resulting sets of core reactions. The authors must resolve the circular argument, since the results obtained are no surprise and represent artefact of the threshold selection approach, and account for the other noted discrepancies.

4. The approach for replacing the biomass reactions must be detailed. The reason is that the authors already comment that forcing flux through the biomass reaction “rescues” some core reactions from central carbon metabolism.

5. Figure S1 cannot be discussed without having access to the size of the models, which are not presented in the present manuscript. Going over the procedure for threshold generation, it seems that the larger number of HK genes may be due to the fact that the extracted models are themselves larger? Another reason why I believe this to be the case is that the comparisons in Fig. 3 A and B are based on fractions, and not the number of reactions!

6. Line 151 – 152 “Overall, we found that StanDep contained more housekeeping reactions than localT2 models (Fig. 3A&B).” –does not—reflect what is shown in the figure panels! These figure compare fractions, which is not the same as numbers. In addition, a larger fraction of HK reactions was found only for mCADRE, MBA, and GIMME (and only MBA and GIMME with respect to HK genes).

7. The higher predictability / accuracy of predictions for essential genes, based on inspection of Fig. S17, is not justified. Further, I do not see what the further grouping of predictions (paragraph on lines 192-200) brings to the performance of the proposed heuristic.

8. The authors talk about “enzyme expression” and “enzyme-specific variability”, which is technically wrong. These can be misinterpreted to mean “abundance of enzyme”, which is not available in transcriptomics data (but only in quantitative proteomics data). This can be corrected by writing “expression of enzyme-coding genes” and “variability of enzyme-coding genes”.

Minor

9. LocalT1, LocalT2, and global should be described for completeness in the main methods, rather than in the supplementary materials.

10. The authors should comment about the reasons for excluding threshold-independent approaches for context-specific network generation.

Reviewer #3: In this manuscript, the authors developed a heuristic method to support the existing model reduction algorithms to prune the genome-scale models to build context-specific metabolic models. The new method takes into account expression pattern across different contexts and determines thresholds based on the hierarchical clustering compared to an arbitrary threshold used in the current methods. The developed method can be useful to improve the quality of context-specific models built from human genome-scale metabolic models; however, such evaluation was only tested purely based on inclusion of housekeeping reactions and gene essentiality. Furthermore, the results are not presented consistently with too many details leaving to the Supplementary material.

Major concerns:

-There are other model reduction methods in the literature that do not use any arbitrary cut off values (e.g., Jensen and Papin, Bioinformatics 2011), so authors need to establish first why to use thresholds after all?

-It is evident from their analysis that the previous methods were not capable of capturing many reactions belonging to housekeeping genes indicating the poor quality of the generated networks. Even with StanDep, mean fraction of housekeeping reactions included in core reaction lists was 0.80, indicating many more housekeeping reactions are still missing in the final models. Therefore, authors need to check the quality of the generated context-specific models based on some functional evaluations together with gene essentiality.

-The authors claim development of hundreds of models for the NCI-60 cancer cell lines that increased inclusion of housekeeping genes. It would be helpful for the reader if you exemplify application of one such model and show how StanDep makes a difference in predicting the outcomes compared to a model developed with arbitrary thresholds.

-The supplementary section contains too many figures, not referred consistently in the main text. Some of those supplementary figures depicting important findings should be in the main text for easy reference to the reader.

- The authors should condense the Supplementary material by reducing the unnecessary figures that are not adding value to the main text.

-As mentioned in the Discussion, the method requires availability of large sets of gene expression data to define thresholds. Therefore, the authors should mention what is the minimum amount of data required to use the method. They should also mention other limitations of their proposed method, if any.

**Have all data underlying the figures and results presented in the manuscript been provided?**

Reviewer #1: Yes

Reviewer #2: No: Large-scale data are used from the original publications.

Reviewer #3: Yes

PLOS authors have the option to publish the peer review history of their article (what does this mean?). If published, this will include your full peer review and any attached files.

Reviewer #1: No

Reviewer #2: Yes: Zoran Nikoloski

Reviewer #3: No

---

## [Decision Letter · Decision Letter 1]

30 Jan 2020

Dear Dr. Lewis,

Thank you very much for submitting your manuscript "StanDep: capturing transcriptomic variability improves context-specific metabolic models" for consideration at PLOS Computational Biology. As with all papers reviewed by the journal, your manuscript was reviewed by members of the editorial board and by several independent reviewers. The reviewers appreciated the attention to an important topic. Based on the reviews, we are likely to accept this manuscript for publication, providing that you modify the manuscript according to the review recommendations.

Sincerely,

Anders Wallqvist

Associate Editor

PLOS Computational Biology

Daniel Beard

Deputy Editor

PLOS Computational Biology

[LINK]

Reviewer's Responses to Questions

**Comments to the Authors:**

Reviewer #1: The authors have adequately addressed my previous concerns. One further suggestion is that all supplemental figures should be mentioned somewhere (main text, Method, or supplemental notes), so that readers have better context when navigating them.

Reviewer #2: Review of the revised version of “StanDep: capturing transcriptomic variability improves context-specific metabolic models”

I would like to thank the authors for attempting to consider all points I raised on the first version of the manuscript. Nevertheless, I find that the revised version still contains some pressing issues which need further explanations. Meanwhile, I found additional issues with the figures in the main text, which I detail in my comments bellow:

1. The authors state that they resolved AND GPR rules, but did not consider OR GPR rules. Integration of transcriptomics data with both rules is well established, as the authors state in their response, so I do not see the reason for neglecting the consideration of OR rules. This must be addressed as the decision seems arbitrary.

2. The presentation of the approach to arrive at the thresholds based on the different clusters is greatly improved. However, what struck my curiosity is that if the respective thresholds \\Theta_c is 100%, then the mean of the data set is used. In how many cases in the analysis did this occur? Again, this sounds like an arbitrary choice!

3. Another point to arbitrariness is the way histograms are created – the number of bins used will certainly affect the derived distance measures. The authors should detail some well-established rules from data science used to address this point.

4. I still do not agree with the justification regarding the number of clusters used. This, too, is an arbitrary choice, since the > 90% similarity is an arbitrary choice. (I do accept the argument about the subsequent Jaccard comparison, to justify the usage of a particular distance measure).

5. The authors should detail the replacing of the biomass reaction, since the methods section should suffice to understand what was done in the present study.

6. The performance of the approach, other than the inclusion of more housekeeping genes (Table 1 and Fig. 3), does not truly provide further conclusive arguments that they improve context-specific models. The authors write that with respect to essentiality they do not perform worse than the other approaches. Hence, more tests are needed to justify the title and this major claim of the manuscript.

7. The authors did not cast their net sufficiently wide to find out that there are other threshold-free approaches for integration of transcriptomics data, other than the one cited. For instance, Robaina-Estevez et al. published and used RegrEx in several situations; comparison with this approach is also desired.

8. The number of supplementary items is staggering. I did attempt to consider all of the tables and figures, but I chose to comment only on the figures of the main text (see next point). The authors may consider removing items which are not truly necessary.

9. In figure 1, it is desirable to somehow specify that distributions are clustered, rather than gene expression profiles (this may be apparent from the y-axis label, but it could be more explicitly stated). Figure 2 A, x-axis label makes no sense; Figure 2B, x-axis label has issues; Figure 3 C – check the % in the pie charts, some do not correspond to the visuals. Figure 3D is very difficult to follow: what are total predictions? Accurate predictions with respect to what? accuracy in % (figure panels should be self-explanatory and understandable based on the caption provided.)

10. I find that one example in Fig. 3C is too little a justification for the success of the presented approach.

Reviewer #3: No further comments

**Have all data underlying the figures and results presented in the manuscript been provided?**

Reviewer #1: Yes

Reviewer #2: Yes

Reviewer #3: Yes

PLOS authors have the option to publish the peer review history of their article (what does this mean?). If published, this will include your full peer review and any attached files.

Reviewer #1: No

Reviewer #2: No

Reviewer #3: No
---

## [Editor Report · Decision Letter 2]

2 Mar 2020

Dear Dr. Lewis,

We are pleased to inform you that your manuscript 'StanDep: capturing transcriptomic variability improves context-specific metabolic models' has been provisionally accepted for publication in PLOS Computational Biology.

Best regards,

Anders Wallqvist

Associate Editor

PLOS Computational Biology

Daniel Beard

Deputy Editor

PLOS Computational Biology

---

## [Editor Report · Acceptance letter]

5 May 2020

PCOMPBIOL-D-19-01550R2 

StanDep: capturing transcriptomic variability improves context-specific metabolic models

Dear Dr Lewis,

I am pleased to inform you that your manuscript has been formally accepted for publication in PLOS Computational Biology. Your manuscript is now with our production department and you will be notified of the publication date in due course.

With kind regards,

Bailey Hanna
